

# Technical Note: Gas Phase Pesticide Measurement Using Iodide Ionization Time-of-Flight Mass Spectrometry

Trey Murschell, S. Ryan Fulgham, Delphine K. Farmer

Department of Chemistry, Colorado State University, Fort Collins, CO, 80523

*Correspondence to*: Delphine K. Farmer (Delphine.Farmer@colostate.edu)

**Abstract:** Volatilization and subsequent processing in the atmosphere is an important environmental pathway for the transport and chemical fate of pesticides. However, these processes remain a particularly poorly understood component of pesticide lifecycles due to analytical challenges. Most pesticide measurements require long (hours to days) sampling times coupled with

off-line analysis, inhibiting observation of meteorologically driven events or investigation of rapid oxidation chemistry. Here, we present chemical ionization time-of-flight mass spectrometry with iodide reagent ions as a fast and sensitive measurement of four current-use pesticides. These semi-volatile pesticides were calibrated with injections of solutions onto a filter and subsequently volatilized to generate gas phase analytes. Trifluralin and atrazine are detected as iodide-molecule adducts, while permethrin and metolachlor are detected as adducts between iodide and fragments of the parent analyte molecule. Limits of

detection (1 second) are 0.37, 0.67, 0.56, and 1.1 µg/m³ for gas phase trifluralin, metolachlor, atrazine and permethrin, respectively. The sensitivities of trifluralin and metolachlor depend on relative humidity, changing as much as 70% and 59%, respectively, as relative humidity of the sample air varies from 0 to 80%. This measurement approach is thus appropriate for laboratory experiments and potentially near-source field measurements.

## 1 Introduction

The widespread use of agricultural pesticides has resulted in the observation of many such compounds in soil and water samples. While the ecological implications of these observations have been the focus of much research, little work has focused on their atmospheric chemistry, despite the fact that many of these pesticides are transported through the atmosphere (Bedos et al., 2006; Coscollà et al., 2008; LeNoir et al., 1999; Majewski et al., 2014; Peck A.M, 2005; Rice et al., 2002; Sauret et al., 2000; Tabor, 1965; White et al., 2006). The chemical fate of compounds in the atmosphere, including oxidation, gas-particle

partitioning, and surface deposition ultimately controls their chemical identities and concentrations, and thus their impact on ecosystems and human health. Global atmospheric concentrations of pesticides are typically considered to be small, yet the local concentrations near point sources can be large enough to result in pesticide drift to neighboring farms, negative impacts on pollinator populations, and substantial occupational exposure of agricultural workers (Choi et al., 2013; Harris et al., 2010).

Trifluralin, atrazine, permethrin, and metolachlor are among the top twenty most frequently used pesticides in the last decade

(Todd et al, 2016). These compounds are thought to be much less toxic, less carcinogenic, and less likely to bio-accumulate than the organochlorine pesticides deemed persistent organic pollutants (POPs) (C. A. DeWit et al., 2004; Clausen et al., 1974; Loomis et al., 2015; Oehme and Ottar, 1984; Shen et al., 2005). However, the environmental and health impacts of current-use pesticides and chemical parameters controlling their atmospheric fate are not well understood. These pesticides have been studied with regard to deposition and volatilization (Glotfelty et al., 1989; Rice et al., 2002), but the investigation of surface-

atmosphere fluxes is limited by currently available instrumentation. In order to fully understand transport, concentrations, and chemical behavior in the atmosphere, *in situ* measurements of these pesticides need to be fast (<1 hour), sensitive and selective in both the gas and particle phases (Farmer et al, 2010).


Volatilization of these current-use pesticides from agricultural soils to the atmosphere removes up to 27% of the applied pesticide (26.5%, 12.4%, and 7.5% for trifluralin, metolachlor and atrazine, respectively) (Rice et al., 2002). The resulting

atmospheric concentrations vary, and these current-use pesticides have been detected hundreds of meters to kilometers away from application sites in both gas and particle phases (Coscollà et al., 2010; Coscollà et al., 2008; LeNoir et al., 1999). Atrazine, trifluralin and metolachlor have been observed near application areas in concentrations ranging from <1 ng/m$^3$ to as high as 61 µg/m$^3$, and in urban and remote locations that are far from sources with concentrations <2 ng/m$^3$ (Foreman et al., 2000; Majewski et al., 2014; Peck A.M, 2005; Bedos et al., 2006; Coscollà et al., 2010).

For detection of pesticides in the atmosphere, air samples are typically collected on solid phase micro-extraction (SPME) fibers or other adsorbent materials with sampling times of 2 h – 1 week (Bedos et al., 2006; Glotfelty et al., 1989; LeNoir et al., 1999; Majewski et al., 2014; Peck A.M, 2005). These solid adsorbents are analyzed by offline techniques, typically gas chromatography coupled to mass spectrometry (GC/MS) or electron capture detection (GC/ECD) (Bedos et al., 2006; Coscollà et al., 2010; Foreman et al., 2000; Glotfelty et al., 1989; LeNoir et al., 1999; Majewski et al., 2014; Peck A.M, 2005; Rice et

al., 2002). These measurement approaches are adequate for offline quantitation of airborne pesticides with detection limits for trifluralin ranging from 1.3 pg/m$^3$ (GC/ MS) to 0.4 µg/m$^3$ (GC/ECD) with sample collection times of 24 hours and 2 hours, respectively (Peck A.M, 2005; Bedos et al., 2006). These techniques have proven successful in quantifying atmospheric pesticide concentrations, although offline analysis introduces steps that can alter a compound's structure and reduce sampling efficiency (Coscollà et al., 2010; LeNoir et al., 1999; Peck A.M, 2005; Rice et al., 2002), and are inadequate for rapid ambient

measurement. Rapid measurements are necessary for (i) observing pesticide drift in real-time to understand meteorological effects; (ii) directly measuring volatilization and surface-atmosphere fluxes by eddy covariance or other micrometeorological approaches; (iii) determining whether agricultural workers are exposed to low concentrations over a long period of time, or high concentrations over a short period of time, and thus for identifying activities that can be targeted to reduce exposure; and (iv) laboratory smog chamber or flow reactor measurements in which oxidation chemistry is typically observed on timescales

of minutes. Studies of oxidation chemistry require such rapid measurement as atmospheric lifetimes of pesticides can be short due to reaction with OH radicals. Atkinson et al. reported rapid trifluralin reaction with the OH radical (>1 x 10$^{-10}$ cm$^3$ molecules$^{-1}$ s$^{-1}$) and photolysis rates on the order of minutes, while the half-life for atrazine plus OH is 2.6 hours for a global average radical concentration of 1 x 10$^6$ molecules cm$^{-3}$ (Atkinson et al., 1999). As an example of the need for rapid detection for indoor exposure estimates, Vesin et al. showed that pesticides must be measured rapidly (<1 h) due to high emission

variation from electronic vaporizers.[44] Here, we investigate the use of chemical ionization mass spectrometry for real-time *in situ* atmospheric pesticide measurement.

Chemical ionization mass spectrometry (CIMS) has been previously explored for pesticide detection. Dougherty et al. detected aromatic chlorinated pesticides, including DDT and DDE (dichlorodiphenyltrichloroethane and dichlorodiphenyldichloroethylene, respectively), using isobutane reagent ions in positive and negative mode (Dougherty et al.,

1975). Tannenbaum employed chloride as a reagent ion to detect the chlorinated pesticide Aldrin (Tannenbaum et al., 1975). More recently, Vesin et al. showed that high-sensitivity proton transfer reaction mass spectrometers can be calibrated to measure indoor concentrations in the 0.5-600 ppb$_v$ range of four pyrethroid pesticides with limits of detection (LODs) of 50 ppt$_v$ with 1 s time resolution (Vesin et al., 2012). However, these studies are limited by the use of a quadrupole mass spectrometer, which only has unit mass resolution (m/Δm ~ 1000), and thus limits the selectivity of the measurement. As

pesticides are typically quite large with molecular weights of 200-500 Da, a separation step before analysis of ambient air is often required to eliminate other isobaric molecules that may act as interferences. CIMS is increasingly used to measure trace gas species in the atmosphere because of high sensitivities, resolution, and selectivities of the different reagent ions employed.





(Bertram et al., 2011; Crounse et al., 2006; Lee et al., 2014; Lopez-Hilfiker et al., 2014; Nowak et al., 2007; Brophy, 2015; Wentzell et al., 2013). Chemical ionization can be achieved by positive or negative ions, including hydronium, acetate, iodide,
and nitrate. CIMS coupled with high resolution time-of-flight mass spectrometry (ToF-MS) is a viable detection technique for atmospheric pesticides because of its fast time resolution (1-10 Hz), field portable design, high mass resolution (m/Δm 4000-6000), and mass accuracy (<20ppm). These features result in measurements of numerous compounds in a complex atmospheric matrix with no need for pre-separation. The elemental composition of analyte ions can be determined for a broad range of *m/z* ratios (typically 0-1000) (Aljawhary et al., 2013, Ehn et al., Lee et al., 2014; Brophy, 2015; de Gouw and Warneke, 2007).
Iodide is an obvious target reagent ion for pesticide CIMS, as iodide has been used to measure oxidized nitrogen and halogenated species, including $N_2O_5$, $ClNO_2$, and $ClNO_3$, and more recently semi-volatile organic compounds, particularly organic acids (Huey et al., 1995; Kercher et al., 2009; Lee et al., 2014). Pesticides often contain one or multiple of these previously detected functional groups, suggesting that iodide is an appropriate reagent ion for their detection.

In this paper, we explore the potential of iodide ToF-CIMS to detect and quantify four current-use, semi-volatile pesticides:
atrazine, metolachlor, permethrin, and trifluralin. We present calibrations using heated injections into an iodide CIMS, and demonstrate that these compounds can be detected at atmospheric and laboratory relevant concentrations with fast (seconds – minutes) time resolution.

## 2   Experimental Method

### 2.1   Chemicals

Standard solutions of trifluralin in acetonitrile (98 ng/μL ± 5%; Sigma Aldrich) and metolachlor in acetonitrile (103 ng/μL ± 5%; Sigma Aldrich), atrazine in methyl tert-butyl ether (MTBE) (1032 μg/mL ± 12; SupelCo, Bellefonte, PA), and permethrin in acetone (999 ± 26 μg/mL; SPEX CertiPrep, Metuchen, NJ) were used in the present study. Solvent choice was dictated by commercial availability of standards.

### 2.2   High Resolution Time-of-Flight Chemical Ionization Mass Spectrometer (ToF-CIMS)

The ToF-CIMS (Tofwerk AG, Switzerland and Aerodyne Research, Inc., Billerica, MA) and iodide (I⁻) ionization scheme used herein are described elsewhere (Lee et al., 2014; Brophy, 2015). Briefly, our Iodide ToF-CIMS has five primary components: the ion molecule reactor (IMR), two RF-only quadrupoles, an ion lens focusing region, and a time-of-flight mass analyzer (m/Δm ~4000) with a pair of microchannel plate detectors. Sample air is continuously drawn into the IMR at 1.9 sLpm (standard Liter per minute) where the sample interacts with iodide reagent ions. I⁻ is generated by flowing ultra-high
purity (UHP) $N_2$ (99.999%, AirGas) over a $CH_3I$ permeation device; the $N_2$ carries gaseous $CH_3I$ into a [210]Po source to produce I⁻ reagent ions (Slusher et al., 2004). Iodide is typically thought to ionize neutral species (M) through a ligand exchange reaction with an iodide-water adduct (R1) (Slusher et al., 2004).

$$[I \cdot H_2O]^- + M \rightarrow [I \cdot M]^- + H_2O \tag{R1}$$

However, deprotonated species have also been observed in ambient measurements (Brophy, 2015), though it remains unclear
whether these species are deprotonated in the initial ionization step or declustered during transmission to the ToF detector.

### 2.3   Heated Pesticide Injections and Calibrations

As most pesticide compounds are commercially available as liquids or solids, calibration of these compounds in the Iodide ToF-CIMS necessitates quantitative conversion of solutions to the gas phase. We developed a heated injection system (Figure


S1) based on work by Lee et al. (2014). We placed a new 2 μm pore Teflon Filter (Pall Life Sciences) for each experiment onto an in-line filter holder (Advantec MFS, Dublin, CA) connected by 13 cm of unreactive ¼" (o.d.) PEEK tubing to the IMR. Pesticide solutions were then injected as liquid samples onto the filter. The filter holder was connected to a 4-way stainless steel union tee (Swagelok). A septum was placed in the second port directly opposite the filter, and a third port, upstream of the filter holder, was connected to a dry UHP zero air flow controlled by two 2000 sccm (standard cubic centimeter per minute) mass flow controllers (MKS, Andover, MA). The air flow was directed through a ½" stainless steel tube packed with cleaned

steel wool, and heated to 200°C by a resistive heating wire on the outside of the tube with a PID temperature controller (Omega, Stamford, CT) attached to a thermocouple located between the wire and tube near the exit of the tube. The heated zero air passed over the filter to volatilize the liquid sample to the gas phase before entering the Iodide ToF-CIMS. The fourth port was open to the room to allow the zero air to overflow the system, exhausting to ambient pressure. The zero air flow was always greater than the inflow of the Iodide ToF-CIMS in order to maintain constant pressure in the IMR and a known flow rate over

the pesticide-containing filter.

For each set of experiments, we injected known volumes (1-6 μL) of solvents (blanks) and commercial standard solutions through the septum onto the filter with a 10 μL syringe (Hamilton, Reno, NV). Following each injection, mass spectra time series (Figure 1) were allowed to return to the same signals as zero air before the next injection was made as can be seen in Figure 1b. Following an injection, pesticide-related peaks rapidly increased, and then decayed in 30-120 minutes. Iodide ToF-

CIMS data were collected at 1 Hz.

### 2.4 Data Analysis

In order to identify peaks in the mass spectrum that changed from blank injections during the pesticide injection experiments, we calculated the signal-to-noise ratio (S/N) for every nominal m/z peak in the mass spectrum. Nominal masses with S/N > 3 during the injection period were identified as potential signals from the pesticide samples, and the high resolution mass spectra

were fit at those m/z ratios to identify the elemental composition of each ion contributing to the signal (Tofware 2.4.3, Figure S2) (Brophy, 2015). The mass spectral peaks identified during the injections corresponded to iodide-molecule adducts ($[I \cdot M]^-$, trifluralin and atrazine) or iodide-molecular fragment adducts ($[I \cdot F]^-$, permethrin and metolachlor). The isotopic patterns for each peak were used to verify our identification of elemental compositions based on the natural abundance of isotopes.

We normalize the signal of each pesticide ion by a ratio of the reagent (defined as the sum of the signals of $I^-$ plus its water

adduct $[I \cdot H_2O]^-$) ion signal during the background signal to the reagent signal at each second of the pesticide injection. This normalization accounts for changes in the reagent ion concentration, and thus ion-molecule collision rates and overall ionization rates, and allows for comparisons across different CIMS instruments with different reagent ion count rates and standardization within a single instrument as the ion source ages. The normalization assumes that variations in the ionization efficiency from temperature or pressure changes are adequately captured by the normalization of the summed reagent ions,

although the humidity-dependence suggests that mechanisms may not be simple, and the assumption should be tested for field conditions.

The start of the injection/desorption period is obvious in the data (Figure 1) with a sharp initial increase in signal above the background (zero air signal) count rate; we define the end of the injection/desorption period as the time at which the mass spectral pesticide signal has returned to within 5% of the background count rate. Extended tailing (1-2 h) indicates slow

volatilization of the liquid sample to the gas phase. The background count rate is determined from a 20-40 minute average and standard deviation of signals detected from UHP zero air, and subtracted from the pesticide mass spectral signals described in



the subsequent analysis. We use the time series of each pesticide-relevant mass spectrometric peak to develop a calibration curve by assuming that the total, integrated signal at a given m/z ratio observed during the injection and subsequent desorption is directly proportional to the known mass of pesticide injected on, and volatilized from, the filter. Thus, the signal collected

at each 1 Hz data point is taken to represent the fractional mass of the pesticide standard injection. That is, if 5% of a single injection's background-subtracted mass spectral signal is observed in one second, that signal corresponds to 5% of the calibrant mass injected on the filter. As the flow rate is constant, this fractional mass can be converted into a mixing ratio (parts per billion by volume, ppb$_v$, liters of pesticide per $10^9$ liters of air), and each 1 Hz data point provides an observed signal for a given mixing ratio. Each injection peak can thus be used to construct a calibration curve and derive the instrument's sensitivity

to the analyte of interest at a given high resolution m/z ratio. Multiple injections allow for the calculation of an average sensitivity for each analyte, and were used to determine average limits of detection (LODs, S/N = 3) from the standard deviation (σ) of the average background count rate of the blanks. This calibration approach assumes that all of the standard solution deposited on the filter is volatilized and that the instrument response is linear over the concentration range of each injection/desorption period.

We note that this calculation differs from the calibration approach described in Lee et al., in which the total summed signal for an injection is divided by the number of molecules injected, and then converted to a mixing ratio using the instrument flow rate at one second (Lee et al., 2014). While that calculation is specific for calibrating collected aerosol samples that are subsequently desorbed from a filter surface, as described by Lee et al. for the Filter Inlet for Gas and AEROsols (FIGAERO) (Lopez-Hilfiker et al., 2014), it does not capture the variation in concentration that occurs on the fast (seconds) timescale of

gas-phase variation and measurement, and would result in an increasing sensitivity with decreased mass spectral averaging times. The calibration approach described herein is thus specific for *gas-phase* calibrations using an injection/desorption technique.

### 2.5 Calibration Technique Comparison

To validate this approach of gas phase calibrations by solution injection and thermal desorption, we compared well-

characterized HR-TOF-CIMS calibrations of formic acid from a home-built permeation tube with injections of a formic acid standard solution (2-4 µL of 10 ng/µL of formic acid in acetone) on the injection/desorption calibration setup described herein. However, as the HPLC-grade solvents contain trace quantities of formic acid, identical volume injections of acetone solvent were necessary to identify and subtract formic signal of the solvent blank from each standard solution injection/desorption period.

### 2.6 Relative Humidity Tests


Due to the ligand-switch mechanism described above, analyte detection by iodide ToF-CIMS is expected to vary with ambient relative humidity (RH) (Kercher et al., 2009; Lee et al., 2014). As field measurements are a desired outcome in the development of a real-time pesticide detector, we investigated the sensitivity of trifluralin and metolachlor with replicate injections of trifluralin (2.4 µL of 98 ng/µL solution) and metolachlor (3 µL of 103 ng/µL solution) over an RH range of 0-80 % (Figure

S5). The RH of zero air was controlled by bubbling zero air (0-2000 sccm) through water (HPLC grade, Sigma Aldrich) and diluting with dry zero air (2000-0 sccm) prior to entering the heated tube and injection region of the calibration apparatus described above. The RH of zero air entering the heated tube was detected by a RH probe/transmitter (Omega HX71, Stamford, CT).



## 3 Results and discussion

### 3.1 Comparison of Calibration Techniques

The results of the formic acid perm tube and injection calibrations are presented in Figure 2. The injection method produces an average sensitivity to formic acid of $3.8 \pm 0.4$ normalized counts $s^{-1}$ $ppt_v^{-1}$, while the permeation tube calibration produces a sensitivity of $3.8 \pm 0.2$ normalized counts $s^{-1}$ $ppt_v^{-1}$. The injection method signal is more variable than the permeation tube, likely due to the large formic acid background in acetone, uncertainties in the syringe volume, and larger, more variable background formic acid in the zero air due to thermal decomposition of species to formic acid in the heated stainless steel tube. Error in the concentration of formic acid from the permeation tube arise from uncertainties in the mass loss rate in the home-built permeation oven. The formic acid injections are much shorter (<1 min) than the pesticide injections, indicative of its higher vapor pressure than the pesticides, but the data analysis is identical. The injection calibration method for CIMS does prove to be sufficient for calibration of lower-volatility compounds and can be used for the pesticides presented herein, for which calibration by permeation tubes is impossible.

### 3.2 Pesticide calibrations

The calibration technique described herein is a dynamic gas phase calibration approach for low volatility compounds that are otherwise challenging to quantitatively convert to the gas phase. While target analytes must be soluble in non-reactive solvents that do not substantially interfere with instrument background or reagent ion concentrations, this versatile technique is a viable alternative to permeation tubes, which require 5-15 mL of pure liquid analyte and can take weeks to months for equilibration and mass loss analysis. Thus, this approach enables calibration of semi-volatile and intermediate-volatility compounds that diffuse through typical Teflon permeation tubes too slowly, or not at all, for detectable mixing ratios and for measurable mass loss (an essential component in determining permeation rates). However, this approach makes three assumptions: (1) volatilization of the pesticide solutions does not cause thermal dissociation or other chemistry of the analyte prior to ionization; (2) the sensitivity of the instrument to the detected pesticide ions is linear across the mixing ratio range created during each injection, and (3) complete volatilization of the pesticide after injection on the filter occurs within the integration time.

We tested the first assumption of negligible thermal chemistry during the volatilization step by varying the temperature of the heated air and testing nitrogen as a carrier gas for the calibrations. We injected the permethrin solutions at two different temperatures, 23°C (unheated) and 200°C (heated) and metolachlor solution at three different temperatures, 23, 100, 200°C. The mass spectra were identical in the unheated and heated injections, albeit over substantially longer timeframes, with permethrin requiring 6-9 hours to return to baseline in the unheated experiment versus 150 minutes or less for the heated system. Metolachlor sensitivity decreased substantially during the injections at 23°C and 100°C, due to the inability of the pesticide to volatilize. Permethrin sensitivity decreased 70% during the room temperature injection, therefore injections at 200°C were pursued. The iodide-molecule fragment adduct was the sole ion observed by the mass spectrometer at both temperatures for permethrin and metolachlor, while the iodide-molecule adduct was not observed. Thus, there is no evidence that the permethrin and metolachlor fragments are produced during the volatilization step, and are thus likely generated in the ion-molecule reaction chamber during ionization. Oxidation of the pesticide standards by $O_2$ in zero air could occur to suppress observed concentrations and thus sensitivity; we note that the sensitivity of iodide ToF-CIMS to trifluralin increased to $250 \pm 20$ $ncps/ppb_v$ when UHP nitrogen was used as the carrier gas and no ambient $O_2$ was present in the calibration system or mass




spectrometer. However, we also note that ion-molecule reactions are altered in the absence of ambient $O_2$, and thus use UHP zero air for all calibrations described herein (Brophy and Farmer, 2016).

We test the second assumption, linearity in instrument response, by examining the sensitivities for different volumes and concentrations for each pesticide standard. These different volumes and concentrations reach different mixing ratio ranges: a non-linear detection response would result in a systematic shift in observed sensitivities as the mixing ratios reached larger

ranges. However, the sensitivities of trifluralin, metolachlor and atrazine are normally distributed around the mean with no observable systematic bias given the concentration ranges (Fig. S3). Thus, the assumption of linear instrument response is justified for these three pesticides in the concentration range used. For example, trifluralin injection volumes of 2 μL (mixing ratio range of 0-10 ppb$_v$) provided the same mean sensitivity (180 ± 20 ncps/ppb$_v$) as larger injection volumes (e.g. 4 μL injection, mixing ratio ranged from 0-60 ppb$_v$ gave a mean sensitivity of 160 ± 30 ncps/ppb$_v$) within the error. This observation

is consistent with calibrations of small acids, including formic acid, acetic, propionic, and nitric acids, which have previously shown linear calibrations using identical Iodide ToF-CIMS (Lee et al., 2014; Aljawhary et al., 2013; Kercher et al., 2009). Permethrin standards are less clear, but do not show a consistent trend between sensitivity and injection volume (Figures S3, S4). To test the third assumption of complete volatilization, we first note that the baseline signal of the detected m/z returned to within 5% of the pre-injection value within 30-120 minutes, suggesting that volatilization was complete before subsequent

injections. Further, replicate injections at multiple volumes are normally distributed for atrazine, metolachlor, and trifluralin, suggesting that uncertainties are random, while incomplete volatilization would likely produce systematic error, and thus non-Gaussian distributions. Permethrin has a lower sensitivity for the 4 μL injections than for the 0.8 or 2 μL injections, consistent with incomplete volatilization at the higher (>15 ppb$_v$) concentration.

The iodide ToF-CIMS detected the four pesticides in the gas phase with sufficient sensitivity for laboratory experiments and

certain field settings (Table 2, Figure 2). Figure 2 shows the calibration curves (red) generated by the single injections according to the calculations described above, with the average calibration curve (black) of the four pesticides studied. In the UHP zero air carrier gas, the average background count rates for the analytes in synthetic air are very low, between 1-7 ncps for the four m/z ratios. LODs for gas phase atrazine, trifluralin, metolachlor, and permethrin are 120±20, 50±30, 110±20, and 150±80 ppt$_v$, which correspond to concentrations of 0.56, 0.37, 0.67, 1.1 μg/m$^3$, respectively and are reported in Table 1. These

concentrations are potentially useful for particle phase measurements by the iodide ToF-CIMS where gases are captured on a thermal denuder and particles are subsequently volatilized either using a heated inlet of filter system (Aljawhary et al., 2013; Lopez-Hilfiker et al., 2014). Trifluralin has been detected in a number of field studies with average ambient gas phase concentrations ranging from 0.228-1.93 ng/m$^3$ and detected concentrations as low as 0.0013 ng/m$^3$ (Coscollà et al., 2010; Peck A.M, 2005). Trifluralin volatilization measured the day of application was as high as 61 μg/m$^3$, decreasing by an order of

magnitude the next day (Bedos et al., 2006). Metolachlor average gas phase concentrations are 0.37-12.74 ng/m$^3$ with the lowest reported concentration of 0.0059 ng/m$^3$ (Peck A.M, 2005; Sadiki and Poissant, 2008). Similarly, atrazine average gas phase concentrations covered a similar order of magnitudes (0.0018-8 ng/m$^3$) (Yao et al., 2007; Peck A.M, 2005). The LODs of the iodide ToF-CIMS suggest that this instrument is appropriate for real-time ambient measurements made near agricultural targets several days after application, but not in remote locations. Further, CIMS has been used in atmospheric chamber

experiments to explore oxidation reactions and mechanisms with starting precursor concentrations between 1-100 ppb$_v$ (Paulot et al., 2009; Ehn et al., 2014; Wyche et al., 2007). The iodide ToF-CIMS is thus suitable for chamber and laboratory experiments of pesticide oxidation chemistry and kinetics.



Relative humidity effects are essential to include RH measurements with ambient field measurements of pesticides, therefore trifluralin and metolachlor were measured at multiple relative humidities (Figure S5). The observed pesticide sensitivities decreased by 70% and 59% between 0 and 80% RH for trifluralin and metolachlor, respectively. Only small changes (8%) were observed in the background signal and noise (5-30%) between 0 and 80% RH. These changes in sensitivity caused the LOD to increase from 108 (50) $ppt_v$ at 0% RH to 421 (110) $ppt_v$ at 80% RH for metolachlor (trifluralin). The decrease in instrument sensitivity with increased relative humidity suggests ionization occurs through a clustering reaction with bare iodide reagent ions, rather than a ligand exchange reaction.

Table 2 compares the iodide ToF-CIMS to previous measurement techniques of the four pesticides. Our work, to the best of our knowledge, is the first online detection and quantification of atrazine, trifluralin, metolachlor, and permethrin. Unlike previous work shown in Table 2, the iodide ToF-CIMS detected the pesticides *in situ* with no collection, extraction, and separation on a gas or liquid chromatograph, enabling rapid (1 Hz) detection. The LODs we report are comparable to those reported by Vesin et al. for on-line, *in situ* measurement of transfluthrin, a pyrethroid compound that is structurally similar to permethrin (Vesin et al., 2013). However, LODs calculated in this work are larger than other techniques and could be improved by longer averaging of sampling time (1-5 minutes).

No significant relationship between decay time of the injection and vapor pressure was found. While iodide ToF-CIMS has been typically used for the measurement and quantification of semi-volatile $C_xH_yO_z$ or small oxidized halogen compounds ($Cl_2$, BrO) (Liao et al., 2014; Huey et al., 1995), atrazine ($C_8H_{14}ClN_5$) is a triazine derived compound with multiple amine groups and a chloride. This suggests that iodide ToF-CIMS might be appropriate for detecting other triazine or organic halide compounds (Liao et al., 2014).

### 3.3 Fragmented Pesticides

While trifluralin and atrazine were detected as quasi-molecular ions with the parent molecule clustered with iodide reagent ions, metolachlor and permethrin were detected as adducts of iodide molecular fragments. Fragmentation is a well-known phenomenon in CIMS, but is not typically thought to dominate mass spectra in atmospheric measurements (Lee et al., 2014). However, previous experiments of pyrethroid compounds similar to permethrin using electron impact ionization showed fragmentation at the ester bond (Vesin et al., 2013), the same bond at which the fragmentation occurs in this study. The intact metolachlor molecule was not detected as a quasi-molecular ion ([I·M]⁻, i.e. as an iodide cluster with the parent molecule at *m/z* 410.79) during either room temperature or heated injections, but instead as a fragment clustered to iodide ([I·$C_{11}H_{14}ClNO$]⁻, at *m/z* 337.98). The observation is consistent with fragmentation at the bond between nitrogen and the second carbon of the methoxypropane group, although we did not observe the corresponding smaller fragment ($C_4H_8O$) as either a bare ion or iodide adduct. Similarly, the intact permethrin-iodide adduct was not detected (*m/z* 518.19). Instead, the dichloro-allyl-cyclopropyl acid fragment is detected clustered with iodide ([I·$C_8H_{10}ClO_2$]⁻, *m/z* 334.91) following fragmentation at the ester bond. We note that none of the four pesticides' fragments or molecular ions were observed unbound to iodide reagent ions, as is occasionally observed for some oxidized organic compounds in other iodide ToF-CIMS (Lee et al., 2014; Brophy, 2015).

### 4. Conclusions

This work demonstrates a calibration technique for semi-volatile compounds newly adapted for ToF-CIMS; while applied here to pesticide measurement with iodide ToF-CIMS, this approach may be used in future studies for quantification of other low-, intermediate- and semi-volatile compounds of atmospheric interest using an array of real-time instruments. These calibrations





demonstrate that iodide ToF-CIMS is sensitive, selective and fast enough for on-line measurements of trifluralin, metolachlor, permethrin, and atrazine in the laboratory in the gas phase. Application to field measurements is more challenging than controlled laboratory conditions for two reasons: (1) changes in relative humidity must be considered for quantitative *in situ* measurements, and (2) the large number of potentially interfering peaks in the mass spectrum can make peak identification

ambiguous. However, while the resolution for the ToF-CIMS might be limited at larger *m/z* ratios, these pesticides provide one particular advantage for detection by mass spectrometry: the presence of halogen and other heteroatoms such as nitrogen, sulfur and phosphorus results in detected ions that have distinct isotopic signatures. The fitting procedures used in the Tofware software package allow for confirmation of peak identify not only by the exact mass of the peak fit, but also by the fit of isotope-containing peaks. Such fitting has allowed for measurement of trace compounds in complex environmental and

laboratory samples (e.g. Lopez-Hilfiker et al., 2014; Lee et al., 2014; Aljawhary et al., 2013; Ehn et al.; 2014; Chhabra et al., 2015). For example, Figure S6 shows potential interferences at the expected *m/z* ratios of the four target pesticides based on previous field campaigns relative to the signal for 1 ppb$_v$ of each pesticide. Potential interferences for metolachlor are minor; potential interferences for trifluralin, atrazine and permethrin are more substantial, but as none of the interfering peaks hold identical halogens, the pesticide isotopes at higher masses can be used to validate the observation (e.g. Figure S2). However,

we acknowledge that mass resolution will be a limiting factor for field measurements of these three pesticides that are far from agrochemical sources. Coupling to an aerosol inlet, including a heated tube (Aljawhary et al., 2013) or filter system (Lopez-Hilfiker et al., 2014), will enable particle-phase measurements in the laboratory and potentially in the field. Such real-time measurements are essential for laboratory kinetic and oxidation product studies to understand the atmospheric fate of pesticides, including oxidation chemistry and secondary organic aerosol (SOA) production, and to better understand regional and global

impacts of these widely used compounds.

**5. Acknowledgements**

This work was funded by the Hermann Frasch Foundation (708-HF12).

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



**Figure 1.** Sample data of injections of trifluralin (a, 1 μL) and sequential metolachlor (b, 2 and 4 μL). The observed mass

spectrometer signals are shown as 1s data time series. Trifluralin is observed at m/z 462.01 indicative of clustering between

the iodide reagent ion, while metolachlor is detected at m/z 337.98, a cluster of a molecular fragment and iodide.



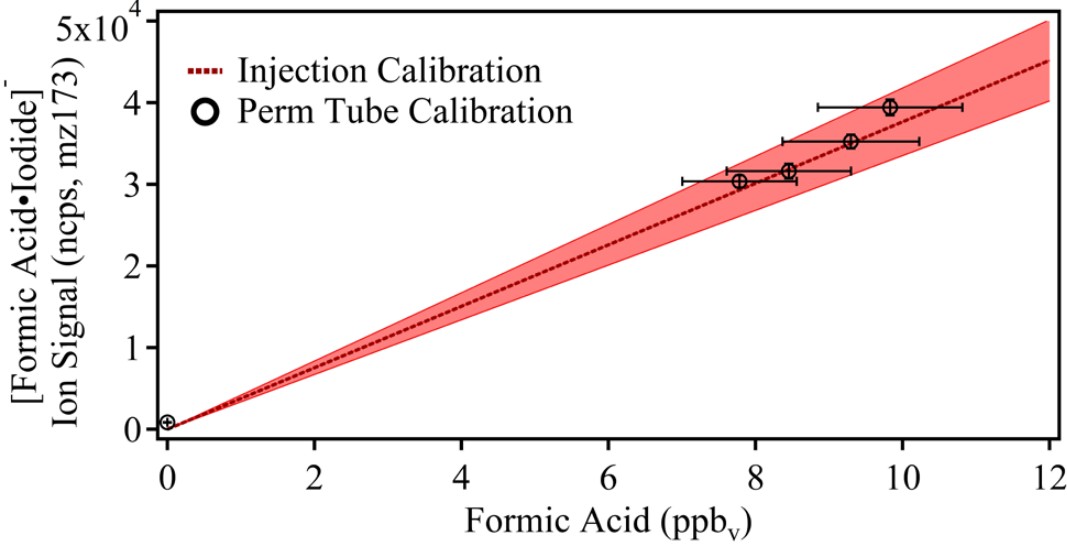

**Figure 2.** Calibration comparison between the conventional formic acid permeation tube calibration method (black open circles) and formic acid injections using the method described herein. Error in the formic acid concentration is represented by the horizontal bars, while error in the y axis is the standard deviation of the signal. The average calibration from the injection method is presented as the dashed red line, and the shaded red area region is the standard deviation of the injection sensitivities.




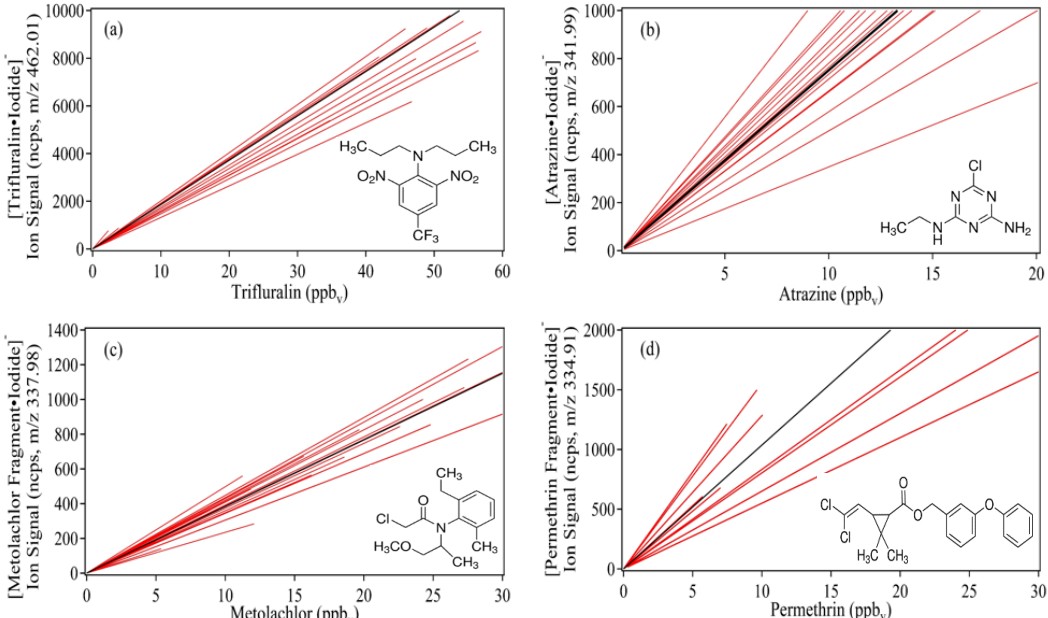

**Figure 3.** Calibration curves of pesticide solutions on the iodide ToF-CIMS, metolachlor (a), trifluralin (b), atrazine (c), and permethrin (d). Red lines represent the signals from single injections as a function of the calculated gas-phase mixing ratio. The average sensitivity (black line) is derived from each set of calibration curves. The error in sensitivity is calculated as the standard error of the average sensitivity for each pesticide.



**Table 1.** Characteristics of the pesticides studied and figures of merit using the Iodide ToF-CIMS

|  | Trifluralin | Atrazine | Metolachlor | Permethrin |
|---|---|---|---|---|
| Pesticide Class | Dinitroaniline | Triazine | Chloroacetanilide | Pyrethroid |
| Use | Herbicide | Herbicide | Herbicide | Herbicide & Insecticide |
| Vapor Pressure (bar) | $6 \times 10^{-8}$ | $4 \times 10^{-10}$ | $2 \times 10^{-8}$ | $2 \times 10^{-11}$ |
| Chemical Formula | $C_{13}H_{16}F_3N_3O_4$ | $C_8H_{14}ClN_5$ | $C_{15}H_{22}ClNO_2$ | $C_{21}H_{20}Cl_2O_3$ |
| Ion Detected | $I^- \cdot C_{13}H_{16}F_3N_3O_4$ | $I^- \cdot C_8H_{14}ClN_5$ | $I^- \cdot C_{11}H_{14}ClNO$ (Fragment) | $I^- \cdot C_8H_{10}Cl_2O_2$ (Fragment) |
| m/z of Ion Detected | 462.01 | 341.99 | 337.98 | 334.91 |
| Standard Concentration | $98 \pm 4.9$ ng/μL | $1032$ μg/mL $\pm 12$ | $103 \pm 5.2$ ng/μL | $999 \pm 26$ μg/mL |
| Solvent | Acetonitrile | Methyl tert-butyl ether | Acetonitrile | Acetone |
| Boiling Point in solution[a] (°C) | 87 | 53 | 81 | 55 |
| Injection Volumes for calibration (μL) | 1,2,3,4 | 1.4, 2.8, 4.5, 6 | 1,2,4,6 | 0.9,1.8, 4 |
| Sensitivity (ncps $ppb_v^{-1}$) | $180 \pm 40$[b] | $75 \pm 19$ | $38 \pm 6$ | $100 \pm 40$ |
| LOD ($ppt_v$)[c] | $50 \pm 30$ | $120 \pm 20$ | $110 \pm 20$ | $150 \pm 80$ |

[a]From manufacturer's data

[b]Error reported as standard error of the average sensitivities

[c]$ppt_v$ = parts per trillion by volume

**Table 2.** Comparison of LODs[a] between iodide ToF-CIMS and previous pesticide measurements

| Reference | Instrument | Approach | Phase | Atrazine | Trifluralin | Metolachlor | Permethrin | Collection Time |
|---|---|---|---|---|---|---|---|---|
| Lenoir et al. 1999 | GC/CIMS[b] | Off-line | Gas | n/a[c] | $1.6 \times 10^{-6}$ μg/m³ | n/a | n/a | >8 h |
| Foreman et al, 2000 | GC/MS-SIM | Off-line | Gas and Particle | $6 \times 10^{-6}$ μg/m³ | $1 \times 10^{-6}$ μg/m³ | $1.2 \times 10^{-5}$ μg/m³ | $2.9 \times 10^{-5}$ μg/m³ | 4 h / 5 min/h |
| Peck et al, 2005 | GC/MS-SIM | Off-line | Gas | $9.8 \times 10^{-6}$ μg/m³ | $1.3 \times 10^{-6}$ μg/m³ | $5.9 \times 10^{-6}$ μg/m³ | n/a | 24 h |
| Bedos et al, 2006 | GC/MS-SIM GC/ECD | Off-line | Gas | n/a | $0.004$ μg/m³ $0.4$ μg/m³ | n/a | n/a | 2-10 h |
| This work | Iodide ToF-CIMS | in situ | Gas | $0.56$ μg/m³ | $0.37$ μg/m³ | $0.67$ μg/m³ | $1.1$ μg/m³ | (No collection) 1 s sampling |

[a]Each work reported different method of calculation for LOD.

[b]Methane chemical ionization mass spectrometry

[c]Pesticide was not studied.