# Peer review of "Technical Note: Gas Phase Pesticide Measurement Using Iodide Ionization Time-of-Flight Mass Spectrometry"

_Atmospheric Measurement Techniques, 2016_

## Referee Comment (RC1) · Anonymous Referee #2 · 15 Mar 2017

Murschell et al. describe the application of iodide ToF-CIMS for on-line detection of four current-use pesticides. They devised a heated injection system to calibrate gas-phase concentrations of the target compounds. The reported sensitivities are sufficient for detecting the target pesticides near application zones and during laboratory studies, but not in remote regions. I think the experimental work was well done and the paper was generally well written. I recommend publication following minor revisions, as suggested below.

Specific comments:

Lines 95-98 and Table 1: Report the standard concentrations as either ng/uL or ug/mL. They are equivalent, so the reason for mixing units is not clear.

[Figure]

Line 154: "...the known mass of pesticide injected on, and volatilized from, the filter." The volatilized mass is not known, it is assumed. Remove that reference from the sentence.

Line 217: "Metolachlor sensitivity decreased substantially during the injections at 23°C and 100°C, due to the inability of the pesticide to volatilize. Permethrin sensitivity decreased 70% during the room temperature injection." Lack of volatilization would change the analyte concentration rather than change conditions that should affect sensitivity (ncps/ppbv). So was there really a change in instrument sensitivity at lower temperatures or was the calculated ppbv merely overestimated by assuming complete volatilization where little occurred? Please clarify.

Line 223: Increased from what value? The measured sensitivities in the presence of O2 have not yet been reported. Would the percent increase be more suitable here?

Line 240: "Further, replicate injections at multiple volumes are normally distributed for atrazine, metolachlor, and trifluralin, suggesting that uncertainties are random, while incomplete volatilization would likely produce systematic error, and thus non- Gaussian distributions." Why would incomplete volatilization lead to non-Gaussian distributions if the resulting mixing ratios were within the linear dynamic range of the instrument? Incomplete volatilization could simply shift the distributions in Fig. S3 and would be difficult to detect without an independent measurement.

Line 263: "Relative humidity effects are essential to include RH measurements with ambient field measurements of pesticides..." This phrase is awkward- reword.

Line 285: Change "...adducts of iodide molecular fragments" to "iodide adducts of molecular fragments".

Section 3.3 should be moved up and likely merged with earlier text (perhaps the paragraph starting on line 212) to remove redundancies. If left as a separate section, it could be better organized.

Fig 3b. Why aren't the endpoints of the low injection volume calibration curves visible like in the other panels? Can the panel be zoomed out? Also, the black curve is thicker than in the other panels.

Showing representative desorption time periods for each pesticide (either in the SI or overlaid in Fig. 1) would help with deciphering the ppbv ranges accessed in each case. The atrazine experiments seem as though they should yield the highest ppbv range (highest mass injected and lowest molecular weight), yet atrazine has the lowest ppbv range in Fig 3. That only makes sense if the desorption rate of atrazine was much slower than the other compounds, but that information is not currently available.

Table 2: Add the "ug/m3" units to the column headings for the four pesticides to reduce clutter within the table.

Technical corrections:

Line 65: Correct the citation format ("44").

Line 191: Change "perm tube" to "permeation tube".

Line 251: Change "of" to "or".

Line 308: Change "identify" to "identity".

Lines 404 and 410: These references are not in alphabetical order.

Fig S2 caption: Change "Each mass spectra are..." to "Each mass spectrum was..."

SI: Under the section 'Relative Humidity Effects' there are several superscripted citations.

SI: The Brophy and Farmer reference lists AMT Discussions as the journal, rather than AMT.

---

## Referee Comment (RC2) · Anonymous Referee #3 · 3 Apr 2017

The authors describe measurements of four commonly used pesticides using chemical ionization time-of-flight mass spectrometry (CIMS), using iodide (water clusters) as chemical reagent ion. The limits of detection of the four pesticides are relatively high, suggesting that the method is adequate for laboratory experiments and, potentially, ambient near-source measurements, but not for more remote measurements. The manuscript is generally well written and the techniques used are sound. I recommend publication of the manuscript after my following comments have been addressed:

Major comments: 1. In section 3.2 the authors list three assumptions made in their measurements and quantification. An additional assumption seems to be that the species (pesticides) are not lost to the inlet or any other internal surfaces of the instrument. I am especially worried about this for the species analyzed as they have low vapor pressure and are therefore more likely lost to surfaces, and request that the authors address this concern in a revised version of the manuscript. 2. An additional challenge to applying the methods to field measurements (discussed in the conclusion) seems to be the limit of detection. This should probably be mentioned again in the conclusion.

Organizational comment: The section on fitting peaks (lines 307-320) seems too much discussion for a conclusion section. I suggest that (most of) this discussion be moved to section 3.

Editorial comment: In line 308 "identify" should be replaced with "identity"

---

## Short Comment (SC1) · 19 Apr 2017

We thank the reviewer for their constructive comments. We have responded to all the minor revisions in a revised manuscript.

Murschell et al. describe the application of iodide ToF-CIMS for on-line detection of four current-use pesticides. They devised a heated injection system to calibrate gas-phase concentrations of the target compounds. The reported sensitivities are sufficient for detecting the target pesticides near application zones and during laboratory studies, but not in remote regions. I think the experimental work was well done and the paper was generally well written. I recommend publication following minor revisions, as suggested below.

[Figure]

Specific comments:

(R.2.1) Lines 95-98 and Table 1: Report the standard concentrations as either ng/uL or ug/mL. They are equivalent, so the reason for mixing units is not clear.

Response: Thank you for catching this; these had been copied from the label from Sigma Aldrich but have been corrected.

(R.2.2) Line 154: ". . .the known mass of pesticide injected on, and volatilized from, the filter." The volatilized mass is not known, it is assumed. Remove that reference from the sentence.

Response: The mass is known from the injection volume and standard concentration, but, as the reviewer correctly points out, the volatilization is assumed to be complete, hence we added "assumed to be completely volatilized" in the revised manuscript.

(R.2.3) Line 217: "Metolachlor sensitivity decreased substantially during the injections at 23_C and 100_C, due to the inability of the pesticide to volatilize. Permethrin sensitivity decreased 70% during the room temperature injection." Lack of volatilization would change the analyte concentration rather than change conditions that should affect sensitivity (ncps/ppbv). So was there really a change in instrument sensitivity at lower temperatures or was the calculated ppbv merely overestimated by assuming complete volatilization where little occurred? Please clarify.

Response: We did assume complete volatilization during the lower temperature injections, and this could be an issue of overestimating how much volatilization took place. However, we do indicate that the signal does return to the baseline albeit after much longer periods of time. We did not see any carry-over of previous injections contributing to signal, no progressively increasing signal in subsequent injections, and no discoloration of the filter after each round of injections.

(R.2.4) Line 223: Increased from what value? The measured sensitivities in the presence of O2 have not yet been reported. Would the percent increase be more suitable

here?

Response: You are correct – this was an artifact of a previous iteration of the manuscript. We report the change to a percent increase in the revised manuscript.

(R.2.5) Line 240: "Further, replicate injections at multiple volumes are normally distributed for atrazine, metolachlor, and trifluralin, suggesting that uncertainties are random, while incomplete volatilization would likely produce systematic error, and thus non- Gaussian distributions." Why would incomplete volatilization lead to non-Gaussian distributions if the resulting mixing ratios were within the linear dynamic range of the instrument? Incomplete volatilization could simply shift the distributions in Fig. S3 and would be difficult to detect without an independent measurement.

Response: The reviewer is correct, incomplete volatilization would shift distributions. The more accurate statement would be "incomplete and inconsistent volatilization", thus producing unpredictable error. We edited the manuscript as such:

"while incomplete and inconsistent volatilization would likely produce unpredictable error, and thus non-Gaussian distributions"

(R.2.6) Line 263: "Relative humidity effects are essential to include RH measurements with ambient field measurements of pesticides. . ." This phrase is awkward- reword.

Response: This was an inaccurately worded phrase. We split the phrase to better say what we intended. That is, we know RH affects sensitivities using iodide CIMS, therefore we tested two of the pesticides. Then, at the end of the paragraph, we affirm that any field measurements with iodide ToF-CIMs need to also include RH measurements. "Due to known relative humidity effects on iodide CIMS sensitivities, trifluralin and metolachlor were measured at multiple relative humidities. . . . Therefore, relative humidity effects on CIMS sensitivity necessitates inclusion of RH measurements with ambient field measurements of pesticides."

(R.2.7) Line 285: Change ". . .adducts of iodide molecular fragments" to "iodide

adducts of molecular fragments".

Response: The reviewer's phrasing makes more sense – we have altered the manuscript. Thank you.

(R.2.8) Section 3.3 should be moved up and likely merged with earlier text (perhaps the paragraph starting on line 212) to remove redundancies. If left as a separate section, it could be better organized.

Response: We moved the beginning of section 3.2 to a new section, 3.4, as it seemed appropriate to introduce the results, then discuss the assumptions to get those results. This also stemmed from the comment on original Line 223, where we present a second result before presenting the first result. After re-reading section 3.3, the reviewer points out a disconnect in the paragraph in which we first discuss pyrethroid compounds detected by mass spectrometry as fragments, then state that we detect Metolachlor as a fragment, before finally stating that permethrin was observed as a fragment. We have re-organized the paragraph to rectify this peculiar paragraph structure. We moved this "Fragmented Pesticides" section into 3.2, before the "assumptions" paragraphs (3.4) to have a better reading progression. We mention in the "assumptions" paragraph that we detect iodide-molecule fragment adducts but decided the fragmented pesticides section is where that should be introduced. We also moved the peak fitting discussion from the conclusion section into the results and discussion section, per another referee's suggestion.

(R.2.9) Fig 3b. Why aren't the endpoints of the low injection volume calibration curves visible like in the other panels? Can the panel be zoomed out? Also, the black curve is thicker than in the other panels.

Response: We have corrected the figure by zooming out the figure and changing the curve line thickness to be more consistent with the other panels.

(R.2.10) Showing representative desorption time periods for each pesticide (either in

the SI or overlaid in Fig. 1) would help with deciphering the ppbv ranges accessed in each case. The atrazine experiments seem as though they should yield the highest ppbv range (highest mass injected and lowest molecular weight), yet atrazine has the lowest ppbv range in Fig 3. That only makes sense if the desorption rate of atrazine was much slower than the other compounds, but that information is not currently available.

Response: The reviewer is absolutely correct. Atrazine and Permethrin standards are an order of magnitude larger than Metolachlor and Trifluralin, thus should have the larger ppbv range. The reviewer is also correct that the atrazine and permethrin desorption periods are much longer. For example, 2 uL injections of Permethrin and Atrazine take 2-3h to return to 5% of the starting baseline, while Figure 1 shows desorption periods of 2uL injections of Trifluralin and Metolachlor take only 20 and 10 minutes, respectively. Atrazine, on average, is longer than permethrin, but within that 2-3 h range. Thus, we added in the methods section 2.4: "Typical desorption periods for injections were 10 min – 1 h for the range of trifluralin and metolachlor injections and 1-3 h for the range of atrazine and permethrin injections." This was added to clarify that the larger standards did in fact have longer desorption periods, thus bringing the ppbv range down. Also, the figure used for the atrazine calibration curve was unnecessarily zoomed in and was not the original figure intended for publication. It was an old figure. This was corrected, depicting the large ppbv range of atrazine detected by the CIMS. This also answers the previous comment's issues.

(R.2.11) Table 2: Add the "ug/m3" units to the column headings for the four pesticides to reduce clutter within the table.

Response: Thank you, this did make the table less messy.

Technical corrections:

Line 65: Correct the citation format ("44").

Response: Corrected, thank you.

Line 191: Change "perm tube" to "permeation tube".

Response: Corrected.

Line 251: Change "of" to "or".

Response: Corrected.

Line 308: Change "identify" to "identity".

Response: Corrected

Lines 404 and 410: These references are not in alphabetical order.

Response: Corrected

Fig S2 caption: Change "Each mass spectra are. . ." to "Each mass spectrum was. . ."

Response: Corrected

SI: Under the section 'Relative Humidity Effects' there are several superscripted citations.

Response: Corrected

SI: The Brophy and Farmer reference lists AMT Discussions as the journal, rather than AMT.

Response: Corrected
* * *

---

## Author Response (AR1)

**Author Response to Anonymous Referee #2**

>*We thank the reviewer for their constructive comments. We have responded to all the minor revisions in a revised manuscript. Please note reviewer comments below, with our responses to each comment indented and in italics.*

Murschell et al. describe the application of iodide ToF-CIMS for on-line detection of four current-use pesticides. They devised a heated injection system to calibrate gas-phase concentrations of the target compounds. The reported sensitivities are sufficient for detecting the target pesticides near application zones and during laboratory studies, but not in remote regions. I think the experimental work was well done and the paperwas generally well written. I recommend publication following minor revisions, as suggested below.

Specific comments:
Lines 95-98 and Table 1: Report the standard concentrations as either ng/uL or ug/mL.
They are equivalent, so the reason for mixing units is not clear.

>*Thank you for catching this; these had been copied from the label from Sigma Aldrich, but have been corrected.*

Line 154: ". . .the known mass of pesticide injected on, and volatilized from, the filter."
The volatilized mass is not known, it is assumed. Remove that reference from the sentence.

>*The mass is known from the injection volume and standard concentration, but, as the reviewer correctly points out, the volatilization is assumed to be complete, hence we added "assumed to be completely volatilized" in the revised manuscript.*

Line 217: "Metolachlor sensitivity decreased substantially during the injections at 23_C and 100_C, due to the inability of the pesticide to volatilize. Permethrin sensitivity decreased 70% during the room temperature injection." Lack of volatilization would change the analyte concentration rather than change conditions that should affect sensitivity (ncps/ppbv). So was there really a change in instrument sensitivity at lower temperatures or was the calculated ppbv merely overestimated by assuming complete volatilization where little occurred? Please clarify.

>*We did assume complete volatilization during the lower temperature injections, and this could be an issue of overestimating how much volatilization took place. However, we do indicate that the signal does return to the baseline albeit after much longer periods of time. We did not see any carry-over of previous injections contributing to signal, no progressively increasing signal in subsequent injections, and no discoloration of the filter after each round of injections.*

Line 223: Increased from what value? The measured sensitivities in the presence of O2 have not yet been reported. Would the percent increase be more suitable here?

>*You are correct – this was an artifact of a previous iteration of the manuscript. We report the change to a percent increase in the revised manuscript.*

Line 240: "Further, replicate injections at multiple volumes are normally distributed for atrazine, metolachlor, and trifluralin, suggesting that uncertainties are random, while incomplete volatilization would likely produce systematic error, and thus non- Gaussian distributions." Why would incomplete volatilization lead to non-Gaussian distributions if the resulting mixing ratios were within the linear dynamic range of the instrument? Incomplete volatilization could simply shift the distributions in Fig. S3 and would be difficult to detect without an independent measurement.

>*The reviewer is correct, incomplete volatilization would shift distributions. The more accurate statement would be "incomplete and inconsistent volatilization", thus producing unpredictable error. We edited the manuscript as such:*

>*"while incomplete and inconsistent volatilization would likely produce unpredictable error, and thus non-Gaussian distributions"*

Line 263: "Relative humidity effects are essential to include RH measurements with ambient field measurements of pesticides. . ." This phrase is awkward- reword.

> *This was an inaccurately worded phrase. We split the phrase to better say what we intended. That is, we know RH affects sensitivities using iodide CIMS, therefore we tested two of the pesticides. Then, at the end of the paragraph, we affirm that any field measurements with iodide ToF-CIMs need to also include RH measurements. "Due to known relative humidity effects on iodide CIMS sensitivities, trifluralin and metolachlor were measured at multiple relative humidities…. Therefore, relative humidity effects on CIMS sensitivity necessitates inclusion of RH measurements with ambient field measurements of pesticides."*

Line 285: Change ". . .adducts of iodide molecular fragments" to "iodide adducts of molecular fragments".
*The reviewer's phrasing makes more sense – we have altered the manuscript. Thank you.*

Section 3.3 should be moved up and likely merged with earlier text (perhaps the paragraph starting on line 212) to remove redundancies. If left as a separate section, it could be better organized.

> *We moved the beginning of section 3.2 to a new section, 3.4, as it seemed appropriate to introduce the results, then discuss the assumptions to get those results. This also stemmed from the comment on original Line 223, where we present a second result before presenting the first result.*

> *After re-reading section 3.3, the reviewer points out a disconnect in the paragraph in which we first discuss pyrethroid compounds detected by mass spectrometry as fragments, then state that we detect Metolachlor as a fragment, before finally stating that permethrin was observed as a fragment. We have re-organized the paragraph to rectify this peculiar paragraph structure. We moved this "Fragmented Pesticides" section into 3.2, before the "assumptions" paragraphs (3.4) to have a better reading progression. We mention in the "assumptions" paragraph that we detect iodide-molecule fragment adducts but decided the fragmented pesticides section is where that should be introduced. We also moved the peak fitting discussion from the conclusion section into the results and discussion section, per another referee's suggestion.*

Fig 3b. Why aren't the endpoints of the low injection volume calibration curves visible like in the other panels? Can the panel be zoomed out? Also, the black curve is thicker than in the other panels.

> *We have corrected the figure by zooming out the figure and changing the curve line thickness to be more consistent with the other panels.*

Showing representative desorption time periods for each pesticide (either in the SI or overlaid in Fig. 1) would help with deciphering the ppbv ranges accessed in each case. The atrazine experiments seem as though they should yield the highest ppbv range (highest mass injected and lowest molecular weight), yet atrazine has the lowest ppbv range in Fig 3. That only makes sense if the desorption rate of atrazine was much slower than the other compounds, but that information is not currently available.

> *The reviewer is absolutely correct. Atrazine and Permethrin standards are an order of magnitude larger than Metolachlor and Trifluralin, thus should have the larger $ppb_v$ range. The reviewer is also correct that the atrazine and permethrin desorption periods are much longer. For example, 2 uL injections of Permethrin and Atrazine take 2-3h to return to 5% of the starting baseline, while Figure 1 shows desorption periods of 2uL injections of Trifluralin and Metolachlor take only 20 and 10 minutes, respectively. Atrazine, on average, is longer than permethrin, but within that 2-3 h range.*

> *Thus, we added in the methods section 2.4 - "Typical desorption periods for injections were 10 min – 1 h for the range of trifluralin and metolachlor injections and 1-3 h for the range of atrazine and permethrin injections."*

*This was added to clarify that the larger standards did in fact have longer desorption periods, thus bringing the ppbv range down.*

*Also, the figure used for the atrazine calibration curve was unnecessarily zoomed in and was not the original figure intended for publication. It was an old figure. This was corrected, depicting the large ppb$_v$ range of atrazine detected by the CIMS. This also answers the previous comment's issues.*

Table 2: Add the "ug/m3" units to the column headings for the four pesticides to reduce clutter within the table.

*Thank you, this did make the table less messy.*

Technical corrections:

Line 65: Correct the citation format ("44").
    *Corrected, thank you.*

Line 191: Change "perm tube" to "permeation tube".
    *Corrected.*

Line 251: Change "of" to "or".
    *Corrected.*

Line 308: Change "identify" to "identity".
    *Corrected*

Lines 404 and 410: These references are not in alphabetical order.
    *Corrected*

Fig S2 caption: Change "Each mass spectra are. . ." to "Each mass spectrum was. . ."
    *Corrected*

SI: Under the section 'Relative Humidity Effects' there are several superscripted citations.
    *Corrected*

SI: The Brophy and Farmer reference lists AMT Discussions as the journal, rather than AMT.
    *Corrected*

**Author response to Anonymous Referee #3**

*We thank the reviewer for the comments - the clarifications have improved the quality of our paper. Our author responses are indented and italicized below.*

The authors describe measurements of four commonly used pesticides using chemical ionization time-of-flight mass spectrometry (CIMS), using iodide (water clusters) as chemical reagent ion. The limits of detection of the four pesticides are relatively high, suggesting that the method is adequate for laboratory experiments and, potentially, ambient near-source measurements, but not for more remote measurements. The manuscript is generally well written and the techniques used are sound. I recommend publication of the manuscript after my following comments have been addressed:

Major comments:

(R3.1) In section 3.2 the authors list three assumptions made in their measurements and quantification. An additional assumption seems to be that the species (pesticides) are not lost to the inlet or any other internal surfaces of the instrument. I am especially worried about this for the species analyzed as they have low vapor pressure and are therefore more likely lost to surfaces, and request that the authors address this concern in a revised version of the manuscript.

> *Response: We used only 13 cm of PEEK tubing from the filter to the entrance of the instrument. Each region in the instrument is under progressively higher vacuum, where contact with surfaces inside the CIMS is minimal to nonexistent.*
> *We added:*
>
> ***"(4) negligible analyte loss to connection between the filter and the CIMS…..***
> ***Finally, to mitigate potential loss of analyte between the filter and instrument, we use the shortest possible piece of unreactive PEEK tube (13 cm) to connect the filter to the iodide ToF-CIMS entrance. There was no evidence of a previously injected pesticide desorbing off the tube wall in ensuing experiments."***

(R.3.2) An additional challenge to applying the methods to field measurements (discussed in the conclusion) seems to be the limit of detection. This should probably be mentioned again in the conclusion.

> *Response: Done.*

Organizational comment:

(R.3.3) The section on fitting peaks (lines 307-320) seems too much discussion for a conclusion section. I suggest that (most of) this discussion be moved to section 3.

> *Response: This comment is consistent with Reviewer 2, and as a result we have moved into section 3.2 where we introduce and discuss the possible use of CIMS for field measurements and discuss the limitations.*

Editorial comment:

(R.3.4) In line 308 "identify" should be replaced with "identity"

*Response: Corrected.*

[revised manuscript text omitted]